# Racial Disparity in Drug Disposition in the Digestive Tract

**DOI:** 10.3390/ijms22031038

**Published:** 2021-01-21

**Authors:** Song Gao, Edward C. Bell, Yun Zhang, Dong Liang

**Affiliations:** Department of Pharmaceutical Science, College of Pharmacy and Health Sciences, Texas Southern University, 3100 Cleburne Street, Houston, TX 77004, USA; Edward.Bell@tsu.edu (E.C.B.); Yun.Zhang@TSU.EDU (Y.Z.); Dong.Liang@tsu.edu (D.L.)

**Keywords:** drug disposition, race, digestive tract

## Abstract

The major determinants of drug or, al bioavailability are absorption and metabolism in the digestive tract. Genetic variations can cause significant differences in transporter and enzyme protein expression and function. The racial distribution of selected efflux transporter (i.e., Pgp, BCRP, MRP2) and metabolism enzyme (i.e., UGT1A1, UGT1A8) single nucleotide polymorphisms (SNPs) that are highly expressed in the digestive tract are reviewed in this paper with emphasis on the allele frequency and the impact on drug absorption, metabolism, and in vivo drug exposure. Additionally, preclinical and clinical models used to study the impact of transporter/enzyme SNPs on protein expression and function are also reviewed. The results showed that allele frequency of the major drug efflux transporters and the major intestinal metabolic enzymes are highly different in different races, leading to different drug disposition and exposure. The conclusion is that genetic polymorphism is frequently observed in different races and the related protein expression and drug absorption/metabolism function and drug in vivo exposure can be significantly affected, resulting in variations in drug response. Basic research on race-dependent drug absorption/metabolism is expected, and FDA regulations of drug dosing adjustment based on racial disparity are suggested.

## 1. Introduction

The gastrointestinal (GI) tract and liver are two major organs of the digestive system. Generally speaking, digesting food ingested from the mouth and absorbing nutrition to support the body are the major functions of the GI tract and the liver. Additionally, the GI tract and the liver can metabolize exogenous toxins and dispose wastes and metabolites to protect the body. For drugs administered through the oral route, which is a preferred route due to many advantages (e.g., safety, patient compliance, ease of ingestion, pain avoidance), the drug molecules must pass through the GI tract and the liver before reaching the systemic circulation and then disease site. In this case, the intestine and liver could be a barrier for drugs because the molecules can be pumped back to the intestinal lumen unchanged or after being metabolized, followed by elimination through feces. This results in low absorption, extensive metabolism, and poor oral bioavailability, which consequentially influence therapeutic efficacy and/or drug safety. Drug absorption and metabolism in the GI tract and the liver are important issues in drug development and in clinical application.

Drug disposition in the GI tract and the liver is affected by many factors, especially metabolic enzymes and drug related transporters. It is well known that metabolic enzymes such as cytochromes P450 (CYPs), UDP-glucuronosyltransferases (UGTs), and sulfotransferases (SULTs) are highly expressed in the liver, catalyzing the metabolism of many drugs [1]. In addition, over recent decades, our knowledge on drug metabolism in the intestine has been significantly expanded, and it turns out that the intestine can also participate in metabolizing many xenobiotics due to highly expressed enzymes such as UGTs and SULTs [2]. Other than metabolic enzymes, drug related uptake and efflux transporters such as P-glycoprotein (Pgp), Breast Cancer Resistance Protein (BCRP), and multidrug resistance-associated protein 2 (MRP2) are also highly expressed in these two organs, which can pump drug molecules and/or their metabolites in or out the body [3,4]. The functions of these enzymes and transporters dominate many drugs’ absorption, distribution, metabolism, and excretion (ADME) and their in vivo exposure [5]. Basic research showed that the expression levels of these enzymes and transporters including the prevalence of their genetic polymorphisms can be highly different in different races at both physiological and pathological conditions [6,7]. The racial differences in drug disposition in the GI tract and the liver can be clinically significant and are worthy of further assessments.

Racial disparity in drug disposition may lead to significant variability in drug exposure variability in different races. For example, it was reported that CYP3A4 activity in Caucasians was different from that in South Asians, resulting in a difference in plasma exposure (AUC) of midazolam when the drug was administered at the same dose [8]. Another example showed that due to higher glucuronidation rates, the clearance of morphine in African-American children was significantly higher than that in Caucasian children, so dose adjustments for morphine was suggested for perioperative patients of different races [9]. Variation of drug related transporters in different races can also affect drug in vivo exposure. For example, it was reported that the systemic exposure of rifampicin was lower in African-Americans when compared to White and Asian populations due to higher frequency of polymorphisms in the SLCO1B1 gene (encoding uptake transporter OATP1B1) [10]. Since interracial variability in PK and in vivo exposure have been observed, which may lead to differences in efficacy and/or toxicity, racially dependent dosing regimens should be recommended to ensure safe and effective use of the drug in specific subpopulations.

Racial disparity in drug disposition and response has been realized, and racial contributions to drug efficacy and safety is an important consideration in drug development. Regulations and guidance regarding racial disparity have been issued or recommended to ensure enrollment of different demographic subgroups in clinical trials [11]. Additionally, race-dependent dosing regimens have been considered and described in the labeling of United States Food and Drug Administration (FDA)-approved drug products as reviewed previously [12]. For example, in the label of tacrolimus, an immunosuppressive drug that prevents rejection after organ transplant, a higher dose in African-American patients is recommended based on the clinical observation of increased metabolism by CYP3A5 in this subpopulation [13]. Interracial variation of drug efficacy/toxicity is important in precision medicine, and regulations have been continuously published. The FDA recognizes the Critical Path Initiative goal to “Continue improving risk-benefit balance of approved drugs by enhancing drug product label language to include pharmacogenetics, where appropriate” [14].

This paper focused on factors that affect drug disposition in the GI tract and liver in different races including interracial variations of metabolic enzymes, transporters, and microflora. In addition, pre-clinical models used to study racial disparity are also discussed. Drug response in different races has been frequently discussed in the literature; however, this was not our focus in this paper since drug response is affected by many factors other than transporter/enzyme function in the GI tract. Many research and review papers have been published on racial disparity in drug disposition, especially drug metabolism in the liver mediated through CYPs. The major SNPs discussed in this paper are listed in Table 1.

## 2. Drug Absorption and Metabolism Disparity in Different Races

### 2.1. Racial Disparity in Drug Absorption

Absorption in the GI tract for orally administered drugs is affected by many factors, among which mucosal permeability is the dominant one. Drug molecules transport across GI epithelia following either passive diffusion or transporter-mediated mechanisms. Passive diffusion permeability is mainly determined by physicochemical properties of drugs. Variation across different populations is not expected to be significant under similar environments. However, when transporters are involved, interracial variation in gut mucosal permeability is significant because protein expression and/or activity can be evidently different, which will alter drug in vivo exposure due to genetic polymorphism in different populations. There are hundreds of membrane transporters in mammalian cells, among which Pgp, MRP2, and BCRP are the major drug related efflux transporters that usually affect drug absorption and bioavailability in the GI tract [15]. We will discuss racial disparity in these three major efflux transporters.

#### 2.1.1. Pgp Polymorphism in Different Races

P-glycoprotein (P-gp), encoded by the *ABCB1* gene and also called Multidrug Resistance Protein 1 (MDR1), is a glycosylated membrane protein containing transmembrane domains and nucleotide-binding domains, which renders the protein as a good carrier for transporting substrates across cell membranes. P-gp was first identified as an efflux transporter in cancer cells where it was found that drugs were pumped out of the cells to cause drug resistance [16]. In the GI tract, P-gp is expressed in the apical side of epithelial cells and participates in pumping drug molecules from blood to the intestinal lumen, resulting in decreased intestinal absorption. Numerous hydrophobic drugs and metabolites such as doxorubicin, etoposide, paclitaxel, SN-38, cyclosporin A, quinine, and digoxin are P-gp substrates; therefore, their absorption in the intestines and their oral bioavailability are highly affected by P-gp. Each year, many papers are published reporting the impact of Pgp on drug absorption, in vivo drug exposure, drug response, and drug resistance.

So far, at least 66 MRD1 SNPs have been identified. The SNPs most widely investigated for their clinical implications are C3435T, C1236T, and G2677A/T. The SNP in exon 26 at position 3435 has been associated with decreased Pgp expression without amino acid changes. The SNP in exon 21 at position 2677 represents an amino acid change of alanine by serine or threonine (Ala893Ser/Thr), as three different nucleotides can be found at this position (G, T, or A). The C1236T in exon 12 that encodes for the TM6 region is essential for substrate binding. For this SNP, there are no amino acid changes. The frequencies of these three SNPs have been investigated extensively. A typical allele frequency in different races is shown in Figure 1. All three alleles, C1236T, G2677T, and C3435T, exhibit the highest frequency in Asian or South Asian populations and lowest frequency among Africans. While C1236T occurs with similar frequency between Asian and South Asian groups, G2677T and C3435T are noticeably more common in South Asians compared to other Asians. Latin Americans that have different ancestries, LA1 and LA2, however, do not exhibit an evident disparity in allele frequencies for all three SNPs.

At a functional level, it is still controversial whether the C3435T allele causes a decrease of Pgp function. Some studies have reported that C3435T deceased Pgp expression in the liver and other tissues, probably due to mRNA stability issues [18,19], while others have failed to confirm the reduction of protein expression [20]. With respect to function, C3435T led to diminished efflux in most in vitro studies [21]. For example, P-gp activity of the efflux of Rh123, a probe of Pgp, was significantly lower in C3435T transfected BLL cells when compared to wild type (3435G) [22]. In in vivo studies, some showed that C3435T decreased drug absorption in the intestine and affected plasma drug exposure. Examples include decreased phenytoin absorption in Indian populations [23], and decreased 3’-p-hydroxypaclitaxel plasma exposure (i.e., Area Under the Curve, AUC) in Japanese ovarian cancer patients with MDR1 C3435T alleles when compared to those of wildtype [24]. However, other studies showed that MDR1 C3435T alleles did not affect the PK profile of certain drugs. For example, it was reported that the PK profiles of fluvastatin, pravastatin, lovastatin, and rosuvastatin were not changed significantly in MDR1 C3435T carriers [25]. These discrepancies are probably caused by variations in drugs used in these different studies, some of which undergo multiple absorption/metabolism pathways. Even though absorption is altered, plasma drug exposure may not be influenced significantly. In addition to efflux function, SNP C3435T can also bring about changes in drug and inhibitor interaction and substrate specificity, probably because this SNP affects the timing of translational folding and insertion of Pgp protein into the membrane [26]. It is intriguing that a synonymous polymorphism like C3435T is able to cause phenotypic effects. One can speculate that epigenetic modifications specific to this SNP may lead to altered concentration of the protein product, which in turn impacts protein functions.

The MDR1 C2677T allele was also reported to be associated with drug resistance and therapeutic outcomes, especially in cancer patients with chemotherapy, probably because of altered drug exposure in the tumor. Similar to the C3435T SNP, pharmaceutical research has shown that the impact of the C2677T allele on Pgp function is inconsistent among different studies. Some studies have shown that C2677T caused a loss of Pgp function, as evidenced by decreased intracellular accumulation and efflux of Rd123, vinblastine, and vincristine in MDR C12677T expressing LLC-PK1 cells when compared to that in wild type cells [27]. This Pgp function loss caused by C2677T was further confirmed by tyrosine kinase inhibitors (e.g., sunitinib, imatinib, nilotinib, dasatinib, and ponatinib) in recombinant cell lines (e.g., Caco-2, K562 c) [28,29]. However, other in vitro studies have suggested that SNP C2677T alleles have no effect on Pgp function in transporting digoxin, verapamil, vinblastine, or cyclosporine A [30]. The impact of C2677T on drug in vivo exposure is more complex because of the interplay between CYPs and Pgp. It was reported that CYP3A4 expression and activity was significantly increased in MDR1 C2677T allele carriers. Increment absorption in the GI tract due to loss of efflux function can be compensated by increased metabolism with CYP3A4 in the liver [31]. In other words, in vivo drug exposure could be decreased even if the absorption in the intestine is increased due to loss of Pgp function.

The impact of C1236T on Pgp expression and function has also been investigated using typical Pgp substrates. In vitro studies using recombinant cell lines showed that C1236T polymorphism decreases P-gp function. For example, the intracellular accumulation of methotrexate, doxorubicin, actinomycin D, and etoposide was higher in C1236T Caco-2 cells when compared the wild type cells [32]. In vivo impact of C1236T was confirmed by meta-analysis from multiple studies, in which it was shown that C1236T impacts drug exposure in the plasma using cyclosporine A as the reference Pgp substrate [33]. However, some in vitro studies showed that C1236T increases efflux function. For example, the efflux ratio (Pba/Pab) of sunitinib, a good Pgp substrate, was 10-fold higher (13.05 ± 0.21 vs. 1.25 ± 0.09 × 10^−6^ cm/s) in C1236T Caco-2 cells when compared to the wild type. One explanation for these inconsistent results is that different compounds at different concentrations were used as the Pgp substrate in these studies.

With respect to racial disparity in drug disposition in the GI tract, South Asian populations were found to have high allele frequency (0.61, 0.65, 0.61) for C3435, C2677T, and C1236T, respectively, while Africans had lower frequency (0.22, 0.12, 0.21) for C3435, C2677T, and C1236T, respectively (Figure 1). Therefore, the expression and/or function of Pgp in South Asian populations are expected to be very different from characteristics observed in other populations.

#### 2.1.2. Breast Cancer Resistance Protein (BCRP) Polymorphism in Different Races

BCRP, encoded by the *ABCG2* gene, was initially identified from a multidrug resistant breast cancer cell line that confers resistance to chemotherapeutic drugs. Then BCRP protein contains six transmembrane domains and functions as a homodimer or homotetramer in transporting drugs out of the cells. Substrates of BCRP include a wide range of chemotherapeutic drugs such as mitoxantrone, camptothecin derivates, flavopiridol, and methotrexate, imatinib, gefitinib, vinblastine, cisplatin, and paclitaxel. Other than drug molecules, BCRP also transports conjugates of certain organic anions, particularly sulfated and glucuronide conjugates such as estrone-3-sulfate, SN-38-sulfate, resveratrol-glucuronides, genistein-glucuronide, etc. BCRP is highly expressed in the GI tract, thus affecting intestinal drug absorption. The highest BCRP expression level in the GI tract is detected in the duodenum, with gradually decreasing expression from the upper to lower parts of the intestine [34].

More than 80 naturally occurring SNPs of the *ABCG2* gene in humans have been identified, among which SNP C421A, C376T, C623T, and G1322A have been studied extensively. C421A is the most well-studied SNP related to drug disposition in the GI tract. This SNP exhibits the highest frequency in Asian populations, and is more common in Latin Americans with mostly European and Native American Ancestry (LA2) than in Latin Americans with Afro-Caribbean ancestry (LA1). In addition, the frequency of the G34A allele is noticeably higher in South Asians than in non-Asian populations (ie., Europeans, Africans, and Latin Americans). In contrast, C421T occurs with a similar frequency among South Asians, Europeans, Africans, and LA1 (Figure 2).

The C421A SNP is associated with lower (30–40%) BCRP protein expression, which was confirmed by western blot analysis in LLC-PK1 cells transfected by plasmid vectors carrying SNP variant cDNAs [35]. Additionally, in vitro studies using different cell lines indicated that the ABCG2 C421A mutation decreased transport function. This was evidenced by lower intracellular accumulation or lower efflux of different BCRP substrates including estrone-3-sulfate, dehydroepiandrosterone-sulfate, *p*-aminohippuric acid, methotrexate, indolocarbazole, gefitinib, erlotinib, and lapatinib when compared to those in wild type cells [35,36,37]. In vitro activity studies also support the above-mentioned function loss. The studies showed that IC_50_ values of BCRP substrates including mitoxantrone, topotecan, SN-38, or diflomotecan against HEK293 cell growth, were higher in C421A cells compared to wild type cells [38]. Mechanism studies showed that the C421A SNP disrupts ATP-binding and therefore impairs energy supply for drug transport. However, one needs to be cautious when concluding that C421 is associated with function loss. When normalized by protein expression levels, the intracellular levels for certain BCRP substrates including estrone-3-sulfate, dehydroepiandrosterone-sulfate, p-aminohippuric acid, methotrexate, and indolocarbazole in C421A transfected cells were similar to those in the wild type cells, suggesting that transport function was not affected [35]. Most in vitro studies only reported intracellular accumulation or transport of BCRP substrates without normalization using protein expression levels, so the notion of function loss due to C421A may need further evaluation.

The AGCG2 C421A SNP has a significant clinical impact. For instance, results from different research groups showed that the pharmacokinetic parameters of rosuvastatin, a robust BCRP substrate, exhibited a significant difference between AGCG2 C421A SNP homozygotes and C421C (wildtype) homozygotes. The plasma Cmax and AUC values of rosuvastatin in C421A homozygotes were up to 2-fold higher when compared to C421C homozygotes, probably because of better intestinal drug absorption due to loss of efflux through BCRP [39,40,41]. The impact of C421A SNP on intestinal absorption was quantitatively analyzed using human PK data for 12 BCRP substrates with a mathematical model. The results showed that in vivo intestinal BCRP transport activity in 421AA homozygous subjects was approximately 23% less than that in 421CC homozygous subjects (wildtype), suggesting function loss with the C421A mutation [42]. However, other research groups reported that this clinical impact is compound dependent. For example, it was shown that C421A SNP affected the pharmacokinetics of fluvastatin and simvastatin lactone, but has little impact on that of pravastatin or simvastatin acid [43]. Another clinical study showed that C421A SNP did not alter the pharmacokinetics of 3H-lamivudine [44].

With respect to racial disparity in drug disposition in the GI tract, Asian populations were found to have the highest allele frequency (0.33) and African populations were found to have the lowest frequency (0.04) for C421A SNP. Therefore, Asian populations are expected to have a higher probability for low BCRP efflux function in the GI tract compared to African populations.

#### 2.1.3. Multidrug resistance-associated protein 2 (MRP2) Polymorphism in Different Races

MRP2, encoded by the *ABCC2* gene and previously named cMOAT, is also a well-studied apical membrane transporter. MPR2 was first cloned from rat liver in 1996, but the function of MRP2 was recognized long before its cloning by studies on the hepatobiliary elimination of organic anions in normal and transport-deficient mutant rats. The substrate spectrum of MRP2 mainly includes anionic conjugates such as bilirubin, bilirubin conjugates, estrone-sulfate, SN-38, SN-38 conjugates, phenolic-glucuronides, phenolic-sulfates, etc.

At least 12 naturally occurring SNPs of the *ABCC2* gene in humans have been identified thus far, among which C24T, G1249A, and C3972T have been studied extensively because of their impact on MRP2 transporter function. In contrast to the above-mentioned ABCB1 and ABCG2 SNPs, the three ABCC2 SNPs—C24T, G1249A, and C3972T—exhibit very diverse patterns of population allele frequency. The C24T allele occurs with the highest frequency among Europeans and Asians, followed by the two Latin American population groups and South Asians. The allele occurs least often in Africans. The G1249A allele occurs most frequently in South Asians and, interestingly, least frequently in all other Asian populations. Its allele frequencies in other populations (i.e., Europeans, Africans and Latin Americans) fall between the allele frequencies in Asian groups. Finally, the order of frequencies of the C3972T allele in different populations—from highest to lowest—is LA1, European, African, LA2, Asian and South Asian (Figure 3).

ABCC2 C24T SNP decreases MRP2 protein expression and function. Genetic analysis using luciferase reporter assays showed that ABCC2 C-24T, both alone and as part of a common haplotype (C-24T/A-1019G/G-1549A), increased promoter function 35% compared to the reference sequence, suggesting MRP2 protein expression may be increased [45]. However, western blot analysis in HEK293T/17 transfected cells showed that the C24T SNP caused about a 27% decrease in MRP2 protein expression [46]. In vivo studies showed that C24T led to lower drug exposure in the plasma. Examples include a 25% increase in plasma MPA-Glu, a potent substrate of MRP2, after a single administration of MPA in Caucasians [47]. In addition, mean plasma methotrexate AUC was up to 2-fold higher in C24T carriers compared to wild types in Caucasian and Chinese individuals [48,49]. Furthermore, the steady state plasma concentration of tenofovir in HIV (human immunodeficiency virus) patients was significantly higher in C24T patients compared to wild type patients (113 ± 47 versus 93 ± 44 ng/mL, *p* = 0.018) [47]. Higher SN-38 plasma AUC was also observed in cancer patients [50]. The plasma AUC of 9-nitrocamptothecin and its 9-aminocamptothecin metabolite was not changed in C24T allele carriers [51]. However, it was reported that these two compounds are substrates of Pgp and are not specific substrates of MRP2 [52]. Since Pgp is also involved in the disposition, the above-mentioned conclusion needs to be examined further.

Contrary to other mutations, G1249A SNP is associated with increased MRP2 protein expression and increased efflux function. In in vitro studies, MRP2 protein expression was increased and intracellular accumulation of multiple MRP2 substrates (e.g., sorafenib, doxorubicin, and paclitaxel) [46,53,54] were decreased using HEK293 and LLC-PK1 cell lines. Consistent with the results from in vitro studies, PK studies showed that the G1249A SNP caused higher intestinal transporter activity, as evidenced by lower oral bioavailability (~20%) and by increased clearance of talinolol when given by i.v. route in German healthy volunteers [55].

Some in vivo studies showed inconsistent results. For instance, a clinical study showed that the absorption rate of mycophenolic acid was about 1.5-fold (C max/AUC, 56% vs. 37%) higher when compared to 1249G carriers [56]. Similar results were found for atorvastatin and cyclosporine in Korean subjects, in which Cmax and AUC of atorvastatin and cyclosporine were significantly higher in 1249A carriers compared to the parameters in 1249G carriers [57,58]. However, the conclusions of these studies need further examination because the patient number was relatively small (n = 30). Another issue is transporter specificity. It is well-known that other than MRP2, cyclosporine is also a good substrate of Pgp. Therefore, the PK profile may not be altered significantly, even if the function of MRP2 is affected.

Although ABCC2 C3972T is also frequently studied, many reports are on drug response resistance, or drug related adverse reactions. Although these effects may be associated with drug absorption and bioavailability, there has been very few direct in vitro or in vivo studies on the impact of ABCC2 C3972T SNP on MRP2 protein expression and function.

With respect to racial disparity in drug disposition in the GI tract, the frequency of these alleles is relatively lower compared to the other two transporters. Therefore, the extent of interracial variations in MRP2 efflux is expected to be less than that of Pgp and BCRP. The C-24T frequency in these populations is very similar, so there is less chance for interracial variation due to MRP2 efflux. In terms of G1249A, which causes high efflux function, South Asian populations have a higher frequency and therefore are expected to have a higher MRP2 efflux in the GI tract.

#### 2.1.4. Genetic Single Nucleotide Polymorphisms (SNPs) and Transport Functions

Many studies have been conducted attempting to correlate the impact of genetic polymorphisms of transporters on drug disposition, especially on drug absorption, PK, and bioavailability. Mathematic algorithms were also developed to predict the impact of transporter/enzyme SNPs on drug disposition [59]. However, results from genetic analysis sometimes does not correlate with those from functional analysis. Sometimes, even controversial or inconsistent results have been observed with the same genetic SNP. For example, as above-mentioned above in Section 2.1.1 (*Pgp polymorphism in different races*), the C2677T SNP of MDR1 caused function loss when tested using tyrosine kinase inhibitors including sunitinib, imatinib, nilotinib, dasatinib, and ponatinib in cell culture models. However, when tested using digoxin, verapamil, vinblastine, or cyclosporine A, this SNP did not cause significant function loss. The mechanism of these observations is not entirely investigated and understood. One of the possible reasons for these inconsistences or controversies is that gene mutation may not necessarily result in protein mutation. It has been reported that the number of SNPs in the human genome is 10–30 million, while the number of protein coding genes in the human genome is only 20,000–35,000, suggesting that genomic SNP may or may not be translated into proteins [60,61]. Another possible reason is that different drugs were used in different studies to evaluate the function. However, these probes may not have the same binding site on the protein. The protein structure of a drug transporter is very complex, which usually include extracellular, intracellular and transmembrane domains. Drug binding sites on the above-mentioned efflux transporters are not completely understood, but different substrates may have different favorite binding sites [62]. For example, it was reported that for the Pgp transporter, at least two binding sites, H-site and R site, were reported and these binding sites have different favorite substrates [63]. Thus, a single mutation may have a different impact on the efflux function if the substrates interact with Pgp at different binding sites. More studies are expected to reveal the relationship across genetic SNP, protein mutation, and transporter function.

### 2.2. Racial Disparity in First-Pass Metabolism

First-pass metabolism, also called first-pass clearance, first-pass effect, or first-pass elimination, is a phenomenon in orally administered drugs where drug molecules are metabolized in the liver or in the intestine before reaching the systemic circulation. The impact of first-pass metabolism on in vivo drug exposure and PK (pharmacokinetics) profiles can be extremely pronounced for orally administered drugs if the drug is a robust substrate of enzymes expressed in the intestine (e.g., UGTs) or in the liver (e.g., CYPs). Many drugs (e.g., 5-fluorouracil, morphine, lorcainide) undergo extensive first-pass metabolism after oral administration [64]. A considerable portion of the drug will be metabolized before entering the blood circulation, and systemic bioavailability is dramatically decreased. The extent of first-pass metabolism in the liver and in the intestine depends on many factors, especially enzyme and transporter expression and activities in the intestine and in the liver. Preclinical and clinical studies have demonstrated that drugs undergoing first-pass metabolism usually have significant interindividual variation in oral bioavailability; therefore, racial disparity in first-pass metabolism should be taken into consideration in clinics because enzyme and transporter expressions could be very different in different races.

#### 2.2.1. Racial Disparity in Cytochrome P450 (CYP) Reactions

Drug metabolism is usually divided into two phases: phase I and II metabolism. Phase I metabolism involves biotransformation such as oxidation, reduction, and hydrolysis catalyzed by different enzymes (e.g., CYPs, ADH, DPD, ALDH). Usually Phase I metabolism increases polarity and water solubility, resulting in either direct elimination through urine and feces or conjugation with other small molecules (e.g., glucuronic acid) prior to elimination. Among the phase I enzymes, cytochrome P450 is the major class including at least 57 functional genes divided into 18 families and 44 subfamilies [65]. CYPs are primarily expressed in hepatocytes involved in the formation or metabolism and breakdown of various endogenous (e.g., cholesterol, fatty acids, lipids) and exogenous (e.g., drugs, dietary components) compounds [66]. It is believed that CYPs are one of the major factors that influence drug in vivo exposure, pharmacokinetics, and drug response. Aside from the liver, CYPs are also distributed in the GI tract [67]. It is reported that among the CYPs isoforms, CYP1A2, 2C8, 2C9, 2C19, 2D6, 3A4, and 3A5 metabolize approximately 90% of current approved drugs (14). Relative and absolute quantitative studies showed that these CYPs isoforms represent approximately 85% of the CYPs in the liver [68].

Genetic variation in CYP genes has been studied in recent decades, with tremendous progress made in human genomic research. Pharmacogenetics, a research area that aims to establish how genetic variation can affect drug disposition and response, has drawn increased attention [69]. SNPs in the CYP gene loci vary among racial groups. These variations can influence the composition of transcribed proteins, leading to variations in gene expression, translation of mRNA, and even activity. It is reported that function loss polymorphisms in CYP genes usually affect splicing and expression rather than transcription or protein structures [70]. The most prevalent alleles of the above CYP isoforms and their clinical impact in different races have been thoroughly reviewed [14,71,72,73,74,75]. The investigations showed that different populations may have allele frequencies that differ greatly from one to another. For example, the frequency of CYP2D6*4 in Asians is 0.4%, as opposed to 28.6% in Caucasians (71-fold). The frequency of CYP2D6*10 is 70% in Asians, and only 6.1% in Caucasians (11-fold). The prevalence of CYP3A5*3 in African-Americans, Caucasians, Asians, and Hispanics is approximately 32%, 93%, 73%, and 65%, respectively. Allele variations cause huge differences in activity in different populations.

Genetic variations can either increase or decrease CYP activity and lead to four types of CYPs including poor metabolizer, intermediate metabolizer, extensive metabolizer, and ultra-rapid metabolizer. Poor metabolizers are homozygous for a variant allele that causes a complete loss of enzyme activity; intermediate metabolizers are heterozygous for a reference allele and a null allele; extensive metabolizers have two active reference alleles; and ultra-rapid metabolizers possess multiple copies of CYP genes. Different levels of metabolizer types have been reported frequently in clinical studies. For example, African-Americans possessing variant CYP3A4*1B alleles (heterozygotes A/G) have about 30% lower midazolam metabolism and lower systemic clearance when compared to GG homozygotes [76]. Another example showed that the efavirenz plasma levels in African-American subjects with the CYP2B6 homozygous 516T/T genotype (reduced function of CYP2B6*6) are approximately 3-fold higher than individuals carrying the homozygous G/G genotype, because of slower metabolism [77].

These phenotypic disparities in CYPs could lead to different drug responses because the clearance and in vivo drug exposure can be significantly different, as above-mentioned. Therefore, it is recommended that dosing regimens require adjustments in certain populations. For example, lipid controlling agents atorvastatin, simvastatin, and lovastatin are good substrates of CYP3A4/5. The recommended daily doses of these agents are 40% greater for intermediate metabolizers (CYP3A5 nonexpressers combined with CYP3A4*1/*1 or CYP3A5 expressers combined with at least one CYP3A4*22 allele) than the doses for poor metabolizers (CYP3A4*1/*22 and CYP3A5 nonexpressers). Similarly, the recommended daily doses for extensive metabolizers (CYP3A4*1/*1 and at least one functional CYP3A5*1 allele) are double the doses for poor metabolizers [78]. Another well-known example is warfarin, an approved anticoagulant commonly used for the treatment of blood clots such as deep vein thrombosis and for the prevention of stroke in high risk people. Application of warfarin in clinics is problematic because of its narrow therapeutic index and significant variability in patient response. Warfarin is a good substrate of CYP2C9, which shows a higher frequency of CYP2C9*2 and *3 in Caucasians (15% and 7.8%) than in African-Americans (1% and 0.5%). These alleles showed only 12% and 5% activity, respectively, compared to the wild type in in vitro studies [79], resulting in higher in vivo exposure in allele carriers. Therefore, the dose of warfarin in these populations should be reconsidered based on the in vivo clearance and racial pharmacodynamic response [80].

It is worth noting that although genetic polymorphisms of CYPs are frequently reported and are relatively easily identified, a disconnect between SNP prevalence and metabolic activity is reported, especially among racial and ethnic groups. For example, CYP1A1 has four common variants including *1, *2, *3, and *4, but only CYP1A1*2C has been associated with activity changes [81].

#### 2.2.2. Racial Disparity in UDP-Glucuronosyltransferases (UGT) Reactions

Phase II metabolism includes conjugation of drug molecules or phase 1 metabolites with endogenous small molecules such as gluconic acid, sulfonic acid, acetic acid, and glutathione to form conjugates with higher polarity to facilitate elimination. Among these reactions, glucuronidation is involved in up to 75% of xenobiotic elimination processes including most clinically used drugs [82]. It is reported that approximately one in 10 of the top 200 prescribed drugs can be eliminated through the glucuronidation metabolic pathway [83]. Since 2012, drug–drug interaction through glucuronidation has been acknowledged in FDA guidelines for industrial drug development.

Glucuronidation is catalyzed by UDP-glucuronosyltransferases (UGTs), which are a superfamily of membrane bound proteins expressed in the endoplasmic reticulum (ER), localized in the luminal side of the ER-membrane. Human UGTs have two families containing at least 22 isoforms, which are mainly expressed in the intestine and liver, but also in other organs (e.g., the kidney) in a tissue-specific manner. For example, UGT1A8 is highly expressed in the small intestine and colon, while UGT1A9 is highly expressed in the liver [84]. Each UGT isoform has its favored substrates, but glucuronidation activity can be compensated across UGT isoforms. Glucuronidation is a major metabolic function occurring in the GI tract and liver to limit the bioavailability of orally administered drugs. For example, raloxifene, an approved estrogen-receptor modulator for the prevention and treatment of osteoporosis in postmenopausal women, undergoes extensive glucuronidation in the intestine, resulting in low systemic exposure [85].

Genetic studies have shown that UGT genes are polymorphic, and more than 200 UGT alleles have been identified in humans [86]. The most prevalent alleles of UGT isoforms in different races have been reported or reviewed previously [87,88,89]. As most previous reviews focused on UGT isoforms expressed in the liver, we will review the racial disparity of UGT1A1 and 1A8 SNPs, two major UGT isoforms highly expressed in the GI tract. UGT1A1 has at least 33 SNPs, among which UGT1A1*6 (211G>A) and *28 ((TA)7TAA) are well studied, probably because these two SNPs are closely associated with drug disposition and response. For UGT1A8, *2 and *3 are major SNPs that are well studied.

Compared with the ABCB1, ABCG2 and ABCC2 SNPs, the frequency of the UGT1A1*6 allele across various populations is low, with 0.155 in the Asian group being the highest prevalence (Figure 4). All other populations including European, African, South Asian, LA1 and LA2 contain this allele with frequencies less than 0.03. UGT1A1*28 occurs with the highest prevalence in African populations and lowest prevalence in Asian populations. The UGT1A8*2 allele occurs with the highest frequency in Asian populations, followed by the European, South Asian, and African populations. The UGT1A8*3 allele exhibits very low rates in all populations studied (Figure 4) and occurs most frequently in Europeans, followed by Africans, and has not been identified in Asian and South Asian groups.

Genetic polymorphism usually leads to abnormal protein expression and glucuronidation activity, aside from altered in vivo drug clearance and exposure. In vitro studies showed that UGT1A1*6 and *28 are associated with lower UGT1A1 expression in human liver microsomes. Furthermore, it was found in an in vitro study that glucuronidation activity in metabolizing estradiol was significantly lower (S_50_, 0.383 ± 0.097 vs. 0.941 ± 0.064 nmol/min/mg) in UGT1A1*28 carriers [90]. Another study showed that when bisphenol A, a good substrate of UGT1A1, was incubated with human liver microsomes, the *Km* was significantly lower using microsomes from UGT1A1*28 carriers (10.6 μM vs. 16.2 μM), suggesting that the binding affinity of bisphenol A with UGT1A1*28 was lower than that of wild type. In addition, the intrinsic clearance (Cl_int_) was lower in UGT1A1*28 microsomes (0.32 mL/min/mg vs 1.24 mL/min/mg), suggesting the loss of glucuronidation function with the *28 SNP [91]. Similar correlations of UGT1A1*28 SNP with glucuronidation activity were also found for bilirubin, ethinyl estradiol, and SN-38 glucuronidation in in vitro studies using human tissue microsomes [92,93,94]. UGT1A1*6 is also associated with lower protein expression and glucuronidation function. For example, when HEK 293 cells were transfected with the UGT1A1*6 plasmid, protein expression levels were significantly lower (~4-fold) than that of the wild type (1A1*1), resulting in lower (~2-fold) SN-38 glucuronidation.

In vivo studies have revealed lower parent drug concentrations or lower glucuronide levels in the plasma of UGT1A1*28 carriers. For example, it was reported that in Japanese patients, the ratio of SN-38-glucuronide/SN-38 in UGT1A1*28 and UGT1A1*6 SNP carriers were significantly lower than in UGT1A1*1 carriers (wild type) (3.65 vs. 6.13 and 4.03 vs. 6.13, respectively), suggesting that these two SNPs decreased UGT1A1 activity and had a significant clinical impact on SN-38 PK [95]. Similarly, other studies have also demonstrated that GT1A1*6 and *28 carriers have a lower plasma ratio of SN-38-glucuronide/SN-38 [95,96]. However, the opposite result was reported in patients treated with raloxifene (60 mg/day for 12 months), in which the raloxifene-glucuronide plasma level was significantly higher (558 ± 115 nM vs. 295 ± 43 nM) in *28 carriers compared to wild type carriers. The authors hypothesized this surprising result was probably caused by enzyme–transporter interplay [97].

For UGT1A8 SNPs, UG1A8*2 and *3, reduced protein expression was demonstrated using transfected Sf9 cells [98]. However, the single amino acid change in UGT1A8*2 (A173G) had little impact on glucuronidation function. In contrast, UGT1A8*3 (C277Y) is associated with a dramatic reduction of glucuronidation function, as demonstrated by mycophenolic acid metabolism in transfected HEK-293 and Sf9 cells [98,99,100]. Similar results were also found using raloxifene as the substrate either using transfected Sf9 cells [98] or using human tissue microsomes from UGT1A8*2 and *3 allele carriers [101]. In vivo studies also showed that UGT1A8*2 had limited impact on mycophenolic acid plasma exposure [102,103].

The most extensively studied drug is arguably irinotecan, a topoisomerase 1 inhibitor used for the treatment of different types of cancers. Irinotecan, a prodrug of SN-38, is a good substrate of UGT1A1. Allele UGT1A1*6 is found to be associated with decreased activity, while 1A1*28, *60, and *93 result in reduced expression. As reviewed previously, these polymorphisms lead to decreased clearance of SN-38, the active and toxic form of irinotecan, which can cause severe drug toxicity including neutropenia and diarrhea. Therefore, toxicity risks among racial groups could differ greatly, so irinotecan dosing regimens are expected to be adjusted accordingly to avoid risk of toxicity. Another well-studied drug is mycophenolic acid, which is a good substrate of UGT1A7/8/9 and UGT2B7. Unlike UGT1A1, some UGT1A7, 1A8, and 1A9 SNP alleles can increase in expression. For example, UGT1A9*1c expresses UGT proteins at a higher level, which may increase glucuronidation activity and result in increased clearance. Therefore, clearance of mycophenolic acid in different races could be increased in the presence of relevant UGT alleles [86].

Other than drug disposition, UGT polymorphism is associated with an increased risk of adverse drug reactions, probably due to variations in drug disposition. It was reported that approximately 10% of the North American population is homozygous for the UGT1A1*28 allele, and these people are at an increased risk of neutropenia with certain drugs. Irinotecan therapy is a prime example, because the clearance of the toxic compound is low in affected patients [104]. The UGT1A1*28 variant is also a common cause of Gilbert Syndrome and Crigler-Najjar Syndrome due to low clearance of bilirubin, a waste product that arises from the destruction of aged or abnormal red blood cells [105]. UGT polymorphism is also reported to be associated with increased cancer risk because of reduced glucuronidation and clearance of endogenous and exogenous carcinogenic compounds [106]. In many cases, multiple UGT polymorphisms are inherited together, and it is often difficult to identify the actual polymorphism that contributes to adverse effects or disease risks. Although UGT polymorphism-associated drug toxicities and disease risks are not well studied, higher risk of drug toxicity and higher disease risks in certain races are predicted through continuous reporting of UGT expression and activity. More attention should be devoted to the prevention of drug toxicity and related disease states in certain races.

## 3. Approaches/Models on Investigating Drug Disposition Disparity

Preclinical and clinical models have been used to study disparity in drug disposition. Each model has its own advantages and disadvantages. In vitro cell culture models, which contain SNPs introduced by transfection, are straightforward. However, cells are absent of biokinetics and lack blood flow, which may not reflect drug disputation in the body. On the other hand, animal models more closely simulate the human scenario compared to cell lines. However, the species differences of transporters and drug-metabolizing enzymes between humans and mice [107] may limit the use of animal models for investigating drug disposition disparity. Therefore, it may be desirable to utilize multiple models to overcome their individual caveat.

### 3.1. Study Impact of SNP on Drug Absorption and Metabolism Using Cells

Transporter or metabolic enzyme SNP transfected cell lines are widely used as in vitro models to determine the impact of SNP on protein expression and transporter or enzyme functions. The HEK293, MDCK, Caco-2, LLC-PK1, K562, Sf9, ES-2, SKOV3, and BLL cell lines have been used previously, most likely because these cell lines can be transfected easily. An important consideration in the selection of cell lines is background protein levels. For example, Caco-2 cells have a high endogenous level of efflux transporters such as Pgp, BCRP, and MRP2, therefore, transfection with Pgp SNP might not be an ideal option due to a high transporter background. Instead, MDCK cells are more suitable for studying Pgp SNPs since they have minimal efflux transporter expression. On the other hand, Caco-2 cells can be used for CYP SNP related studies, as few CYP enzymes are expressed in Caco-2 cells naturally. Another example is HEK293 cells, which have a low level of endogenous UGT protein. HEK293 cells can be used for UGT polymorphism studies with SNP allele transfection so that the glucuronidation function will not be affected by other isoforms.

Specific substrates, usually small molecules, are used in the above-mentioned in vitro models to evaluate the transporter functions. For example, tyrosine kinase inhibitors such as sunitinib, imatinib, nilotinib, dasatinib, and ponatinib have been used to investigate the impact of Pgp polymorphism on efflux function because these drugs are robust substrates of Pgp [108]. Since different transporters—BCRP and MRP2, for example—can share the same substrate, substrate specificity is very important. Results can be misinterpreted if multiple transporters are involved. In addition, since some SNPs express their protein products at a lower level compared to the wild type, it may be necessary to normalize intracellular drug accumulation or drug metabolism rate with the protein level to accurately evaluate protein function. For example, ABCG2 C421A transfected cells exhibited higher intracellular drug accumulation when compared to wild type cells in the above-mentioned studies [35,36,37]. Higher intracellular accumulation usually suggests loss of efflux function. However, when intracellular drug levels were normalized with BCRP protein levels, there was no difference between these two types of cells, suggesting that efflux function was normal and the difference in intracellular drug accumulation was due to different BCRP protein expression levels [35]. Therefore, the conclusions could be inaccurate without normalizing protein expressions.

### 3.2. Study Impact of SNP on Drug Absorption and Metabolism Using Animal Models

In contrast to cell lines, animal models are extremely useful tools to understand the impact of drug transporter and metabolizing enzyme SNPs on drug absorption and metabolism in a whole-body system. Hereto, a number of mouse models carrying one or more of these SNPs have been developed. For instance, a humanized transgenic UGT1A-SNP mouse model was established almost a decade ago, which harbors the Gilbert Syndrome-associated UGT1A variant haplotype including UGT1A1*28, UGT1A3-66T>C, UGT1A6*2a, and UGT1A7*3 [109]. Compared to the corresponding wild type, this UGT1A haplotype led to lower UGT1A mRNA expression and UGT1A protein synthesis. Furthermore, UGT1A transcriptional activation by dioxin, phenobarbital, and endotoxin was significantly reduced in SNP mice. In recent years, CYP3A-humanized mice carrying the *CYP3A5* SNP, CYP3A5*3, have been established. This allele results in the almost absence of CYP3A5 protein expression in the liver and intestine. CRISPR/Cas9 mediated editing of the CYP3A5*3 allele allowed the creation of the CYP3A5*1 SNP mice that recapitulate the *CYP3A5*1* carrier phenotype in humans [110,111]. With the development of the genome editing technology, more animal models carrying different drug transporter or metabolizing enzyme SNPs are anticipated. These models will undoubtedly advance our understanding of the roles of these SNPs in drug disposition in vivo. On the other hand, it is noteworthy that although laboratory animals share many physiological similarities with humans, there are remarkable species differences of transporters and drug-metabolizing enzymes between humans and mice. For example, while mice have at least six Cyp2d (Cyp2d9, 2d 10, 2d11, 2d12, 2d13, 2d22), humans express only one functional CYP2D enzyme, CYP2D6. The activity and substrate specificity of human and mouse CYP2D enzymes are also diverse [112]. In addition, although UGT1A enzymes are the major contributors to hepatic mycophenolic acid metabolism in both humans and rats, 1A9 is dominant in human and 1A1 and 1A7 are likely the principal mediators in rat [113]. The above-described humanized transgenic mice can overcome the caveats of lacking certain enzyme/transporter in mice. However, for some enzyme/transport, their proper functioning may require specific cellular machinery, which might also exhibit differences between species. It is important to be aware of the species difference in drug transport and metabolism, since it represents a major confounding factor in animal data interpretation during drug discovery.

### 3.3. Study Impact of SNP on Drug Metabolism Using Microsomes

Tissue microsomes prepared from SNP allele carriers are a common in vitro model to study the impact of SNP on metabolic enzyme functions. Usually, kinetic parameters (e.g., Km Vmax, Cl_int_) are calculated to fully evaluate substrate-enzyme binding affinity and metabolic activity. Tissues (e.g., liver tissue, intestinal tissue) are collected from patients carrying different SNP alleles to prepare microsomes. Specific substrates are incubated with these microsomes to determine metabolic functions. The results help determine the impact of SNP on the function of enzymes responsible for drug metabolism. For example, SN-38 was incubated with liver microsomes prepared from patients with UGT1A1*6, and *28. Metabolic rates at different SN-38 concentrations were then measured, and kinetic parameters were calculated to determine the impact of *6 and *28 on glucuronidation [90].

### 3.4. Clinical Studies

In clinical studies, specific substrate drugs are usually used to determine the impact of SNP on drug disposition. Thereafter, plasma PK profiles are determined and PK parameters (AUC, t_1/2_, clearance) are calculated. By comparing PK parameters between different SNP carriers, the extent of disparity can be identified. For example, to examine the effects of SNPs on Pgp function, phenytoin and 3’- p-hydroxypaclitaxel were administered to Indian and Japanese populations. PK profiles were obtained, and the results showed that wild type patients displayed significantly higher plasma drug exposure than MDR1 C3435T carriers, suggesting higher oral bioavailability of these drugs due to loss of Pgp efflux functions [23,24]. Aside from plasma drug exposure level, the ratio of metabolite vs. parent compound concentration can be used to evaluate metabolic enzyme function. For example, the ratio of SN-38-glucuronide to SN-38 in plasma demonstrated loss of glucuronidation function in patients with UGT1A1*6 and *28 SNPs [95, 96].

Although racial variations of metabolic enzymes at the genetic level have been frequently reported (Figure 1 and Figure 2), the significance of their clinical impact across races remains uncertain. One reason is that aside from genetic factors, non-genetic factors can also affect the expression and function of an enzyme. For example, females express CYP3A4 2-fold more than males, which corresponds to a 50% incremental change in verapamil metabolism [114]. Another source of uncertainty are compensatory mechanisms. Numerous investigations have shown that many drugs are metabolized by more than one isoform. When the function of one isoform is reduced, others may compensate. This compensation may be seen in mycophenolic acid, an immune-suppressant medication used to prevent rejection following organ transplantation and to treat Crohn’s disease. Mycophenolic acid is a good substrate of UGT1A7, 1A8, 1A9, and 2B7. When the expression and/or activity of one isoform decreases, the drug can still undergo glucuronidation by other enzymes. In vivo exposure may or may not change, depending on both genetic and non-genetic factors. In summary, whether and how racial disparity in drug metabolism in the GI tract and liver impacts in vivo exposure and drug response need to be carefully evaluated.

## 4. United States Food and Drug Administration (FDA) Oversight of Drug Applications Regarding Race and Ethnicity

Despite wide acknowledgement about the importance of genetic variation to successful drug therapy, a lack of studies exist that sufficiently evaluate the effects of pharmacogenomics on health disparities. Poor implementation of pharmacogenomics into clinical practice has been cited as responsible for this reality, due to concerns about its cost-effectiveness, and even due to poor reporting about the impact of pharmacogenomics on disparity. The United States Food and Drug Administration has placed increased emphasis on the issue of racial and ethnic disparity in drug disposition and metabolism in recent years. The agency has adopted the strategy of addressing these disparities through improvements in the conduct of clinical trials [115]. To that end, the FDA offers guidance pertaining to the enrollment in and design of trials that may feature diverse populations, in addition to the collection and handling of data from diverse groups.

### 4.1. Enrollment and Trial Design Recommendations

The FDA promotes the use of reasonably broad eligibility criteria to conduct clinical trials that include participant groups that best represent the populations most likely to use the drugs or biological products pending approval. The agency also acknowledges the common, but often unnecessary exclusion of trial participants based on language barriers or scheduling difficulties. Such exclusions may often result in the unintended effect of making trials less diverse from the standpoint of participant racial and/or ethnic makeup. The FDA recommends that upon greater understanding of pathways of metabolism and excretion for drug approval candidates, expansion of participant eligibility criteria should be considered for future trials. In the case of unnecessary exclusion resulting from historical or habitual practices, the agency recommends similar expansion of eligibility criteria, citing use of the Phase 3 clinical trials as an opportunity for inclusion of these people [116].

More inclusive trial practices are also encouraged. For instance, the FDA recommends consideration of whether Phase 2 clinical trial exclusions may be modified or eliminated in advance of a Phase 3 trial conduct. Exclusion criteria modification may also occur in the conduct of trials focused on higher risk populations. In some instances, such populations are racially or ethnically diverse.

Characterization of drug metabolism and excretion early during the clinical assessment process is encouraged to avoid unnecessary exclusion of diverse populations. In addition, adaptive trial designs are recommended that would allow for the alteration of trial populations while studies are in progress. The FDA also recommends the use of targeted inclusion of specific population groups who might possess a particular disease severity, and for whom a drug’s effect might be more readily demonstrated. The agency also suggests re-enrollment of early-phase clinical trial participants into later-phase trials to assist in obtaining information about the treatment of rare conditions in certain segments of the population [116].

### 4.2. Diminishment of Participation Barriers

The FDA recommends steps be taken to make enrollment processes easier for potential participants, which includes technology-mediated methods for interfacing with participants. In addition, personal interaction including the use of mobile medical personnel may increase enrollment in underrepresented populations. Effective advertisement of financial reimbursements and/or incentives are also recommended to better attract underrepresented groups for study participation [116]. Use of inclusive outreach to the public in advance of clinical trial commencement is recommended. This includes direct engagement with communities, with emphasis on building relationships with potential participants and community leaders and support personnel. Furthermore, post-trial engagement is encouraged to maintain rapport and trust with diverse populations [109]. Locations possessing a higher concentration of underrepresented groups are recommended as sites for conducting inclusive clinical trials. With respect to participant recruitment, use of health providers and study coordinators with shared cultural backgrounds with study participants is a favored strategy. The FDA also encourages frequent scheduling of recruitment events, even in non-clinical locations and during cultural and social gatherings. In addition, the use of multilingual trial brochures and support staff is encouraged to attract participants who have communication limitations [116].

### 4.3. Racial and Ethnic Data Collection

The FDA sets minimum standards for collecting, maintaining, and reporting data pertaining to racial and ethnic groups. The agency recommends the use of a two-question format for the collection of information. The first question should pertain to ethnicity, whereas the second question should pertain to race. Clinical trial participants are recommended to self-report ethnic and/or racial identity. Ethnic identifiers should include the minimum options of “Hispanic or Latino” or “Not Hispanic or Latino”, accompanied by a brief description of the terms. Racial categories are recommended to include, at minimum, identifiers and descriptions including “American Indian or Alaska Native”, “Asian”, “Black or African-American”, “Native Hawaiian or Other Pacific Islander” or “White”. Selection options of more than one racial designation or more subgroup designations are encouraged [11].

The FDA recommends that the number of self-reported respondents in each racial category who choose Hispanic or Latino be reported. In addition, the number of participants who identified only one racial category should be reported. Detailed response distributions possessing all combinations of multiple-race identifications should also be reported, with the minimum expectation that the number of participants identifying more than one race be communicated [11].

## 5. Summary and Perspective

Racial disparity of drug disposition in the GI tract is a very important issue in drug development and clinical practice. Genetic polymorphism of drug-related transporters and metabolic enzymes vary substantially in different races. In vitro and in vivo studies have demonstrated that genetic polymorphisms in transporters and enzymes cause significant differences in protein expression and functions in SNP allele carriers, resulting in highly variable drug absorption in the GI tract and drug metabolism in the GI tract and/or the liver. Many studies have shown that several-fold differences in the plasma exposure of certain drugs occur in various SNP allele carriers. These variations will eventually affect drug response, toxicity, and resistance. Dose adjustment based on SNP mutations has been suggested, and toxicity risk disclosures for certain genetic alleles (e.g., UGT1A1*28) for certain drugs (e.g., irinotecan) are required to be included on drug labels by the FDA. However, since most of these studies are focused on SNP alleles, direct racial disparities in drug disposition in the GI tract and the liver have not been extensively studied. Most of the studies have been based on genetic analysis, but not racial or ethnic variants, which is a direction upon which precision medicine may focus its future. With respect to study models, compared to cell lines, in vivo animal models are superior in determining the impact of genetic variation on drug disposition in the GI tract and in the liver. However, species difference in drug deposition warrants caution in interpreting the animal data.

## Figures and Tables

**Figure 1 ijms-22-01038-f001:**
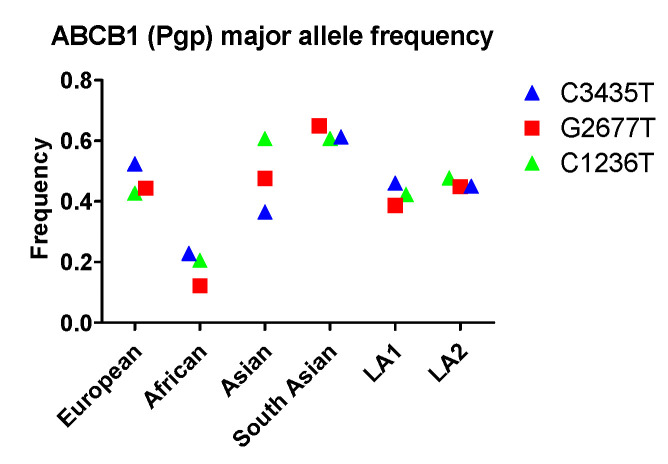
The allele frequency of ABCB1 in different populations. Asian: All Asian individuals excluding South Asians; LA1, Latin American individuals with Afro-Caribbean ancestry; LA2: Latin American individuals with mostly European and Native American Ancestry. Data were extracted from NCBI (National Center for Biotechnology Information) ALFA project (version: 20200227123210, www.ncbi.nlm.nih.gov/snp/docs/gsr/alfa/ [17]).

**Figure 2 ijms-22-01038-f002:**
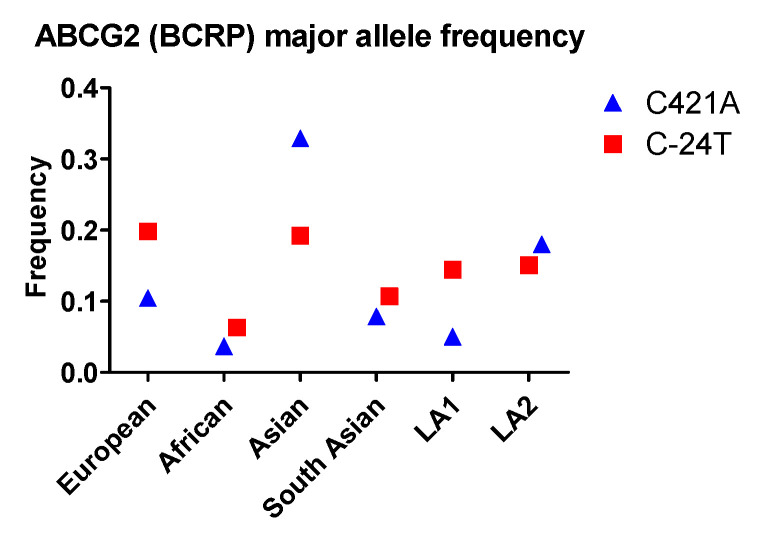
The allele frequency of ABCG2 in different populations. Asian: All Asian individuals excluding South Asians; LA1: Latin American individuals with Afro-Caribbean ancestry; LA2: Latin American individuals with mostly European and Native American Ancestry. Data were extracted from NCBI ALFA project (version: 20200227123210, www.ncbi.nlm.nih.gov/snp/docs/gsr/alfa/ [17]).

**Figure 3 ijms-22-01038-f003:**
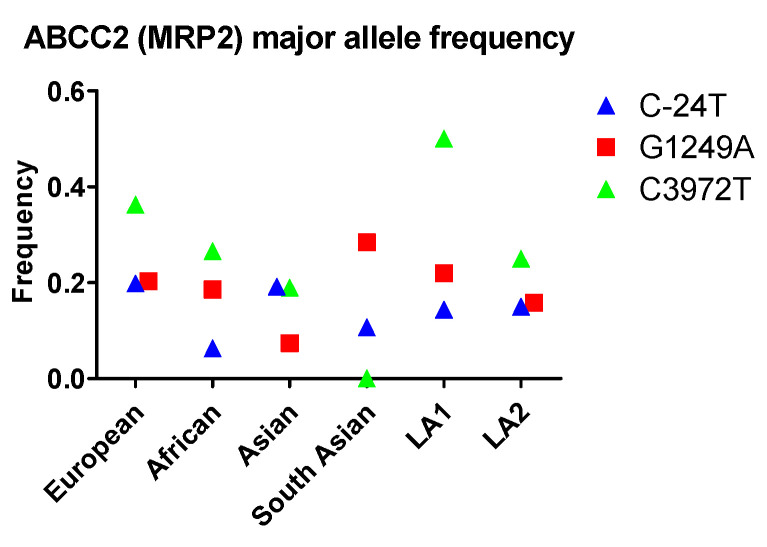
The allele frequency of ABCC2 in different populations. Asian: All Asian individuals excluding South Asians; LA1: Latin American individuals with Afro-Caribbean ancestry; LA2: Latin American individuals with mostly European and Native American Ancestry. Data were extracted from NCBI ALFA project (version: 20200227123210, www.ncbi.nlm.nih.gov/snp/docs/gsr/alfa/ [17]).

**Figure 4 ijms-22-01038-f004:**
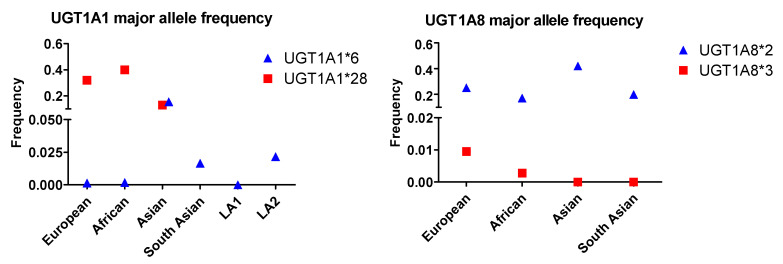
The allele frequency of two UGT1A1 and UGT1A8 SNPs in different populations. Asian: All Asian individuals excluding South Asians; LA1,:Latin American individuals with Afro-Caribbean ancestry; LA2: Latin American individuals with mostly European and Native American Ancestry. Data for the UGT1A1*6 and UGT1A1*28 alleles were extracted from NCBI ALFA project (version: 20200227123210, www.ncbi.nlm.nih.gov/snp/docs/gsr/alfa/ [17]).

**Table 1 ijms-22-01038-t001:** Major transporters and metabolic enzymes single nucleotide polymorphisms (SNPs).

Gene	Protein	Allele	Nucleotide Change	Amino Acid Change	Location	RS Number
ABCB1	Pgp	C1236T	C1236T	G412G	Exon 12	1128503
ABCB1	Pgp	G2677T/A	G2677T/A	A893S/T	Exon 21	2032582
ABCB1	Pgp	C3435T	C3435T	I1145I	Exon 26	1045642
ABCG2	BCRP	C421A	C421A	Q141K	Exon 5	2231142
ABCC2	MRP2	C–24 T	C-24T	-	Promoter	717620
ABCC2	MRP2	G1249A	G1249A	V417I	Exon 10	2273697
ABCC2	MRP2	C3972T	C3972T	I1324I	Exon 28	3740066
UGT1A1	UGT1A1	UGT1A1*6	G211A	G71R	Exon 1	4148323
UGT1A1	UGT1A1	UGT1A1*28	TA repeats	-	Promoter	3064744
UGT1A8	UGT1A8	UGT1A8*2	C518G	A173G	Exon 1	1042597
UGT1A8	UGT1A8	UGT1A8*3	G830A	C227Y	Exon 1	17863762

## Data Availability

Not applicable.

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
