# Peer review of "Racial Disparity in Drug Disposition in the Digestive Tract"

_ijms, 2021, doi:10.3390/ijms22031038_

Round 1

Reviewer 1 Report

This is a very well researched review which, when published, could be a useful resource material for a broad description of current understanding regarding the role of genetics of race in drug disposition. One limitation, which the authors mentioned, is the fact that most of the work done in this field deals with genetics of oncology and related drugs. The authors presented the collated data in clear and concise manner. Tables & figures are appropriate. I recommend this article for publication after a thorough proof-reading. 

Proof-reading of this manuscript is needed before publication as in several instances spacing between two words has been omitted. For example, P1, line 12 (‘withemphasis onthe’), P2 line 63 (‘forperiooperative’), P2 line 69 (‘beenobserved’) and P17, line 798 (‘focusits’). These need to be corrected. 

The 4th section (page 15, line 649) titled “FDA regulation of drug application regarding to races/ethnicity” has succinctly presented some essential regulatory information which could be useful for advanced researchers however, most of this information presented here could be accessed from the FDA website where greater details regarding FDA’s rules & regulations could be found. Therefore, inclusion of this section only moderately adds to the importance of this review. 

Author Response

We apologize for the misspelling/lack spaces in the text. We have carefully checked the entire manuscript and correct those errors.

We agree the comment on section 4. We have shorten this section dramatically including remove the "Enrollment recommendations (4.2)" and "Racial and Ethnic Data in Labeling Procedures (4.5)". However, we would like to keep the other sections because these sections may be important to disparity related research. 

Reviewer 2 Report

The review entitled "Racial disparity in drug disposition in the digestive tract" is very clear and detailed. I am not a specialist about SNP and this review has brought me many informations.

Comments :

Concerning drug transporters, it is well known that transporters have several binding site for different substrates. Also, an SNP can modulate partially the activity of one transporter only for one substrate but not necessarily for the others. This notion lacks especially for MDR1.

It lacks spaces, line 69 (beenobserved), 332 (ofPgP), 492 (oppositeresult), 553 (idealoption), 612 (evaluatesubstrate), 693 (recommendsuse).

Author Response

We appreciate the commons and we added a new section "2.1.1 Genetic SNP and transport functions".

We apologize for the misspelling/lack spaces in the text. We have carefully checked the entire manuscript and correct those errors.